# Emerging Preclinical Applications of Humanized Mouse Models in the Discovery and Validation of Novel Immunotherapeutics and Their Mechanisms of Action for Improved Cancer Treatment

**DOI:** 10.3390/pharmaceutics15061600

**Published:** 2023-05-26

**Authors:** Isha Karnik, Zhisheng Her, Shu Hui Neo, Wai Nam Liu, Qingfeng Chen

**Affiliations:** 1Institute of Molecular and Cell Biology, Agency for Science, Technology and Research (A*STAR), 61 Biopolis Drive, Proteos, Singapore 138673, Singapore; isha_karnik_from.tp@imcb.a-star.edu.sg (I.K.);; 2Department of Microbiology and Immunology, Yong Loo Lin School of Medicine, National University of Singapore, Singapore 117593, Singapore; 3Singapore Immunology Network, Agency for Science, Technology and Research (A*STAR), 8A Biomedical Grove, Immunos, Singapore 138648, Singapore

**Keywords:** humanized mouse model, cancer, immunotherapy, drug development, drug repurposing, drug discovery, disease mechanisms, resistance mechanisms

## Abstract

Cancer therapeutics have undergone immense research over the past decade. While chemotherapies remain the mainstay treatments for many cancers, the advent of new molecular techniques has opened doors for more targeted modalities towards cancer cells. Although immune checkpoint inhibitors (ICIs) have demonstrated therapeutic efficacy in treating cancer, adverse side effects related to excessive inflammation are often reported. There is a lack of clinically relevant animal models to probe the human immune response towards ICI-based interventions. Humanized mouse models have emerged as valuable tools for pre-clinical research to evaluate the efficacy and safety of immunotherapy. This review focuses on the establishment of humanized mouse models, highlighting the challenges and recent advances in these models for targeted drug discovery and the validation of therapeutic strategies in cancer treatment. Furthermore, the potential of these models in the process of uncovering novel disease mechanisms is discussed.

## 1. Introduction

Immunotherapies are considered the new ground-breaking strategy in cancer treatments. Cancer immunotherapy aims to reinvigorate the immune system, which is suppressed by tumour cells, and enhances the anticancer effects of cytotoxic immune cells. Studies on immunotherapies originated from understanding the mechanisms of tumour escape in the process of tumour development by targeting the pathways and mechanisms involved, which include the manipulation of cytokine expression, regulation of immune checkpoint receptors, and expansion of immune-modulatory cells. These studies have led to more than 3000 different kinds of cancer immunotherapies being studied or entering development, some of which include therapies based on immune checkpoint inhibitors (ICIs), chimeric antigen receptor (CAR)-based (CAR-T cells, CAR-natural killer (NK) cells) therapy, oncolytic virus- and cancer vaccine-based therapies, and combinatorial therapy [1]. Despite the hype surrounding these immunotherapies, there is a strong need to improve their translation into clinical practice, reduce their toxicities, and overcome the resistances of some cancers. While experimental results have been promising, their translation into clinical practice has been sub-optimal.

The targeted inactivation of inhibitory immune receptors has showed promising results and suggested that immunotherapy can potentiate antitumour responses in advanced cancers [2]. This was initially observed in patients with advanced melanoma [3] and subsequently extended to patients with other types of cancers. However, the clinical response rate has been unsatisfactory in patients [4], likely due to the species-specificity of immunotherapy drugs. This could lead researchers to move into clinics without sufficient data on safe and effective starting doses, relying mainly on historical, theoretical, and in vitro data, which would lead to severe adverse effects for patients who enter into clinical trials.

Although studies conducted in humans would be the most relevant for in vivo research, there are certain ethical issues which make this approach unfeasible. This fact has brought comparative medicine, wherein the similarities and differences between veterinary medicine and human medicine are evaluated to draw preclinical inferences, to the forefront of research [5]. A major part of our understanding of various biological processes has originated from animal studies, especially those involving rodent models. Their ease of maintenance and genetic manipulation, short reproductive cycle, and similarities in genomic, anatomical, and physiological characteristics with humans have made rodents widely used in mammalian model systems for in vivo studies [6]. They provide exceptional experimentally tractable model systems, with 80% of mouse genes being proportional to human orthologs, that can help researchers examine essential molecular mechanisms of diseases and treatment responses [7].

Mouse models gained importance in cancer research for preclinical studies about 50 years ago with the introduction of tumour transplantation in wild-type mice [8]. These models, which allowed for the rapid affordable testing of potential cancers, remain commonly used in vivo cancer tumour models due to their reproducibility [8]. Historically, in order to study human cancers more precisely, immunodeficient or immunocompromised mice transplanted with patient-derived xenografts (PDXs) derived from fresh human tumour biopsies, or engrafted with human cancer cell lines, were used for a wide range of studies as they were affordable and were believed to present significant predictive insights into clinical outcomes when reviewing the abilities of new cancer therapeutics [8]. The discovery of the CRISPR/Cas9 system further revolutionised the field of transgenic mice models by generating efficient mice models with higher rates of knock-in and knock-out success and gene editing [9]. This technology has demonstrated great potential to generate a wide spectrum of mutations found in human tumours, thereby offering rapid cancer modelling in mice and providing novel solutions [10,11]. For instance, the use of transgenic mice has greatly improved drug screening and reduced drug toxicity, leading to the development of successful drug compounds [11].

Despite these advantages, the CRISPR/Cas9-based technology is limited in validating the oncogenic potential of putative oncogenes. It may also cause the activation of the immune system against Cas9-expressing cells, leading to their clearance along with creating unsuitable off-target mutations in the utilised models [9,10]. Substantial lengths of time and resources are required to generate these transgenic mice, alongside the challenge of overcoming heterogeneity within mouse phenotypes [9]. The presence of stromal murine cells and the lack of a human immune system in these models remain major challenges, leading to high false-positive error rates that impede therapeutic results and fail to demonstrate the intricate relationship between adaptive and innate immune systems and tumour microenvironments (TMEs). Moreover, several elements of the mouse immune system are highly distinct from those of the human immune system. Differences in innate molecules, like the lack of toll-like receptor (TLR) 10 in mice and the expression of TLR11, TLR12, and TLR13 (which are absent in humans) in mice, limit the use of mouse models for creating ideal comparisons [12].

The discrepancies between the findings from experiments involving wild-type mice and their translations into clinical trials have led to high failure rates in clinical trials, as seen in the Phase-1 trial of TGN1412, a novel agonist anti-CD28 monoclonal antibody that stimulates T cells [13]. While preclinical murine models showed no detectable toxicity, the clinical trial resulted in a systemic inflammatory response characterised by cytokine storm, multiorgan failure, and severe depletion of lymphocytes and monocytes, thus having severe life-threatening consequences [13]. Similarly, while the experimental studies and early preclinical effects of Odronextamab (REGN1979), a human CD20 × CD3 bispecific antibody from Regeneron, demonstrated positive antitumour activity [14], this antibody was later linked to severe consequences and two deaths due to cytokine release syndrome (CRS) in patients [15]. Additionally, even with the promising nature of CAR-T cell therapy, which has had successful responses in haematological malignancies, it has met with similar unexpected and life-threatening immune modalities like CRS, neurotoxicities, and even fatality, despite passing the safety dose testing in wild-type mice [16].

There is enough evidence to note the lack of research predicting immunotoxicities in wild-type mice, emphasizing an urgent need to establish a reliable platform to evaluate the safety and efficacy of preclinical immunotherapeutic drugs and predict patient response with an accurate TME representation. This realisation has led to the development of humanized mouse models. These models have played a massive role in the field of oncology in recent years, especially with the rise of immunotherapeutics. They have since evolved from being platforms for drug testing to being fundamental components in the process of drug discovery, development of new targeted therapies, and discovery of novel disease mechanisms.

This review focuses on the impact that humanized mouse models have had on the many breakthroughs in biomedical science through their application in various studies on human diseases, like cancer, and immunology. Moreover, it highlights the potential benefits of leveraging these models, explores the uses of these models in discovering novel disease mechanisms and drug discovery, and addresses efforts made to improve these models for greater utility in translational research.

## 2. Applications of Humanized Mice in Oncology

Most cancers are classified into immune-hot and immune-cold groups based on their immune infiltrations, with immunotherapy showing more significant effects on the supposedly immune-hot groups as seen in the case of melanoma [17]. Despite the safety of some drugs in certain cancers, they may cause adverse reactions or unsuccessful treatment in others. For example, lung airways or gut-associated lymphoid tissues have different immune environments from those of the liver or pancreas; this could be a reason for the failure of targeted therapies [18]. A well-developed humanized mouse model can overcome these limitations and serve as a preclinical platform for testing the safety and efficacy of new and existing drugs, along with characterising adverse events and identifying the best sequence of immunotherapy agents for therapy. Using a humanized mouse model allows researchers to test the antitumour cytotoxic responses of immune cells and assess the safety of a drug in the presence of both tumours and immune systems. Performing a pharmacokinetic (PK) and/or bioavailability study in humanized mice over time, and with different dosages, can help clinical scientists evaluate the stability of a given drug in patients and safely identify an evidence-based dosing regimen for that drug [19]. As different patients may respond differently to the same drug, using peripheral blood mononuclear cells (PBMCs) from patients that are to be treated could be used to predict the reactions of these patients to the drug, thus presenting a path to personalised medicine.

Essentially, a humanized mouse model is one that has human immune cells engrafted into immunodeficient mice. There are three commonly known methods of generating these humanized models (Figure 1), which include inoculation with PBMCs, inoculation with CD34+ hematopoietic stem cells (HSCs) alone, or simultaneously engraftment with human foetal liver and thymic tissue (BLT) into the sub-renal capsules of immunocompromised mice. Engraftment with purified human PBMCs via intraperitoneal injection into homozygous severe combined immunodeficient model (SCID) mice was first performed in 1988 [20]. Over time, immunodeficient models were improved (Table 1), and advanced NSG (Il2rgnull) mice, which show a high chimerism rate, the development of human erythropoiesis and thrombopoiesis, and the formation of haemato-lymphoid compartments, have become the bases for the development of today’s humanized mice models [21,22].

The possibility of PBMCs and tumour tissues originating from the same patients proved to be a key advantage of the Hu-PBMC model, but resulted in only a partial reconstitution of the human immune system in the peripheral blood of the mice, with most cells being CD3+ T cells that eventually interacted with major histocompatibility complex (MHC) molecules in mice and led to graft-versus-host disease (GvHD) [20]. This incomplete reconstitution could be improved via the engraftment of HSCs characterised by the expression of the cell surface marker CD34, which is obtained from umbilical cord blood, peripheral blood, or the foetal liver. These models showed higher leukocyte reconstitution and greater engraftment efficacy than other sources and provided insights into the organisation of the human stem-cell hierarchy and bases for functional assays, which had been difficult to achieve with the earlier models [35]. However, their use was impeded as mice from the NOD/SCID strain appeared to develop a limited number of B and T cells, in a phenomenon called leakiness, during aging [36]. These mice also had higher levels of host murine NK cells, with significantly low human NK cell counts—possibly to compensate for the lack of lymphocytes—which eventually hampered the engraftment of human cells [35]. Human T cells undergo the process of human leukocyte antigen (HLA)-restriction and T cell selection in the thymus, which enables the T-cell receptor (TCR) to recognise and bind to certain MHC molecules that present foreign antigens and initiate a T-cell immune response. NSG mice, however, express the H2 complex, not HLA molecules, on their thymic epithelial cells, and thus the T cells in the HSC model lack HLA-restriction. This can be overcome with the advancement of the BLT mouse model, allowing for T cell maturation in an autologous human thymus, overcoming the murine H2 restriction, and promoting human T cells that are capable of HLA-restricted antigen-specific reactions [37,38]. Despite these ideal conditions, the BLT model is technically challenging to develop and may lead to immunologic rejection due to the existence of mismatched HLA between immune cells and tumour tissues. The discussed models also lack human NK cells, and the need for foetal tissue has raised several ethical issues.

A broad spectrum of specific humanized oncology models has been generated, including those for human nasopharyngeal cancer (NPC) [39], hepatocellular carcinoma (HCC) [40], gastric carcinoma [41], colon carcinoma [41], prostate cancer [42], bladder cancer [43], head and neck squamous cell carcinoma (HNSCC) [44], lymphoma [45], and melanoma [46,47]. These can be either cell-line-derived (CDX) or PDX models. Our lab has been at the forefront of establishing cancer-specific humanized mouse models in Southeast Asia and has demonstrated the importance of specificity in these models, pointing out that the immune system can have different effects on the TMEs of different cancers. For example, the growth of the tumour in the NPC humanized model in the presence of the human immune system is significantly inhibited, whereas in the HCC model, the size and weight of tumours are increased as compared to what one would find in the immunodeficient mice [39,40].

Humanized mouse models allow detailed studies of the TME and tumour cells’ interactions with the human immune system, providing deeper insight into the mechanisms of tumour progression, metastasis, and development of immunotherapies. They have been used in studies on different immunotherapies including antibody-based ICI, adoptive cellular therapy (ACT, which encapsulates methods like CAR-T and CAR-NK cell therapy), cytokine manipulation, cancer vaccination, and the development of oncolytic viruses as elaborated in this review (Figure 2).

### 2.1. Humanized Mouse Models for Testing ICIs and Antibody-Based Drugs

Increasingly, evidence of dysregulated levels of immune checkpoints in tumour-infiltrated T cells has been noted, thus posing these cells as attractive targets for immunotherapies. These checkpoints include cytotoxic T-lymphocyte antigen 4 (CTLA-4), programmed cell death protein 1 (PD-1), lymphocyte activation gene 3 (LAG-3), and T-cell immunoglobulin and mucin domain 3 (TIM-3), as seen in Figure 3 [48]. 

ICIs form a part of monoclonal antibody (mAb) targeted therapy. Humanized mouse models play an important role in the preclinical study and testing of ICIs and their interactions with tumour-infiltrating lymphocytes (TILs), thus helping in predicting the efficacy of these drugs as seen in Table 2. Here, we discuss some of the models that have provided new insight into their respective studies and have helped pave the way for further discoveries.

The HCC and NPC humanized mouse models established in our lab were able to successfully validate existing anticancer and combinatorial therapies [39,40,49]. These models successfully showed that cancer cells educate the immune cells and condition the TME to suppress the immune system and aid in tumorigenesis. The HCC model also replicated the immunotoxicity caused by ipilimumab (anti-CTLA-4 antibody) in patients [40], while an exploration of the NPC model highlighted similarities between clinical data from patients and the results from this model [39,50]. Both models indicated the presence of an exhaustive phenotype [50] with the presence of TILs and the upregulation of inhibitory receptors, thus establishing a human-specific platform for drug testing and a path to using these models for discovering new therapeutic strategies and drug combinations.

Further, a PDX humanized model for non-small cell lung cancer (NSCLC) has indicated the hCD8 T-cell-mediated efficacy of pembrolizumab (anti-PD-1 antibody) and shown the significant growth inhibitory effect of pembrolizumab on these tumours [51]. Similarly, a sarcoma PDX humanized mouse model showed consistently elevated levels of hCD8+ T cells and their subsets, implying the effect of these cells in antitumour activity, with reductions in tumour size [52]. This model also recapitulated the sarcoma TME as seen in clinical findings and predicted the NK-related factors that affect the survival of patients [52]. These findings show the importance of these models in verifying cellular and molecular targets for drug development and hint towards the mechanisms associated with their actions.

**Table 2 pharmaceutics-15-01600-t002:** Safety and efficacy profiles of a few ICI-based therapies. A brief description of some cancers that were modelled in humanized mice and used for testing the safety and efficacy of human-specific ICI-based drugs.

Mouse model	Target	Safety	Efficacy	Model Description	Refs.
Colon	CD137 and PD-1	√	√	CDX-model with intraperitoneal injection of PBMCs in Rag2^−/−^IL2Rγnull strain	[41]
Gastric	CD137 and PD-1	√	√	PDX-model with intraperitoneal injection of PBMCs in Rag2^−/−^IL2Rγnull strain	[41]
HCC ^1^	PD-1 and CTLA-4	√	√	PDX-model with intrahepatic injection of human CD34+ HSCs in NSG mice	[40,49]
NPC ^2^	PD-1 and CTLA-4	√	√	PDX-model with intrahepatic injection of human CD34+ HSCs in NSG mice	[39]
Lymphoma	PD-1 and CTLA-4	√	√	CDX-model with subcutaneous engraftment of human PBMCs in NOG mice	[53,54]
Sarcoma	PD-1	√	√	PDX-model with intravenous injection of human CD34+ CB cells in NSG mice	[52]
NSCLC ^3^	PD-1	√	√	CDX and PDX-models with intravenous injection of human CD34+ HPSCs in NSG mice	[51,55]
	PD-L1	√	√	CDX and PDX-models with intravenous injection of either PBMCs or human CD34+ HSCs in NSG mice	[56]
Bladder	PD-1	√	√	PDX-model with intravenous injection of human CD34+ HPSCs in NSG mice	[51]
TNBC ^4^	PD-1	√	√	CDX and PDX-models with intravenous injection of human CD34+ HPSCs in NSG mice	[51]
Lung adenocarcinoma	PD-1	√	√	CDX-model with intravenous injection of CD34+ human HSCs in NOG and NOG-FcγR^−/−^ mice	[57]
	PD-L1	√	√	CDX-model with intravenous injection of cord-blood derived CD34+ human HSCs in NSG mice	[58]
HNSCC ^5^	PD-1	√	√	CDX-model with intravenous injection of CD34+ human HSCs in NOG and NOG-FcγR^−/−^ mice	[57]
Ovarian carcinoma	PD-L1	√	√	CDX-model with intravenous injection of fetal liver-derived CD34+ human HSCs in NOG mice	[58]

^1^ HCC Hepatocellular Carcinoma. ^2^ NPC Nasopharyngeal Carcinoma. ^3^ NSCLC Non-Small Cell Lung Cancer. ^4^ TNBC Triple-Negative Breast Cancer. ^5^ HNSCC Head and Neck Squamous Cell Carcinoma.

### 2.2. Safety and Efficacy Profiling of ACT in Humanized Mouse Models

With the hype surrounding personalised medicine, ACT has gained much importance with respect to cancer immunotherapy. It has indeed shown promising results, especially with respect to metastatic melanoma [59] and B-cell malignancies [60]. This method involves identifying and isolating immune cells or lymphocytes from a patient, expanding them ex vivo, and then infusing them back into the cancer patient, targeting the tumour cells, and resulting in antitumour activity. However, treatment-related toxicities have limited their widespread implementation in therapeutic regimes, with experimental studies being irreproducible in patients. With the use of humanized mouse models, these challenges are being addressed with enhanced treatment efficacy prediction.

#### 2.2.1. CAR-T Cells

CARs are constructed by integrating the single-chain fragment variable (scFv) domain of an antibody with the TCR constant domain, presenting the TCRs with high affinity and specificity for target antigens. This allows the activation of modified T cells in an MHC-independent manner, thus bypassing the challenge posed by tumour escape [61]. CAR-T therapy, especially when involving anti CD19-CAR-T cells, has shown impressive results in clinical trials involving B-cell malignancies [62,63] and multiple myeloma [64]. However, only approximately half the patients in these trials were shown to respond, while some acquired resistance and suffered major toxicities including CRS, neurologic toxicity, and B-cell aplasia, thereby limiting the potential of this therapy [60]. A humanized mouse model could be used not only to design innovative CAR constructs and develop novel CAR-T cell therapies but also to evaluate their safety and efficacy against disease.

Considering the severe toxicities of CRS in ACT, murine models mimicking CRS in patients have been developed to generate a better understanding of CRS pathophysiology and thereby improve the safety, and reduce the toxicities, of CAR-T therapies. A study using SCID-beige mice to model CRS reported serum cytokine levels that were similar to those reported in clinical studies and established the important role of macrophage-derived IL-6, IL-1, and nitric oxide (NO) in the pathophysiology of CRS [65]. Genetically engineered CAR-T cells that constitutively produce IL-1 receptor antagonist were shown to protect from CRS-mortality along with maintaining the antitumour efficacy in NSG mice; this finding is useful when attempting to identify a new target for CRS and also to design a novel CAR construct [65]. This genetic engineering technology can be boosted by silencing unwanted or faulty genes to provide optimal therapeutic effects. This technique was recently employed to silence PD-1 expression in CAR-T cells and has showed improved polyfunctionality of CAR-T in vitro and in PDX models [66]. The functionality of silencing PD-1 and gene editing techniques can be further investigated in humanized mouse models. As such, gene-edited CAR-T cells with disrupted PD-1 signalling were shown to have enhanced activity in humanized CDX-derived glioblastoma mouse models [67]. The utilised method prompts the development of allogeneic CAR-T cells to provide off-the-shelf therapy rather than relying on autologous CAR-T cells.

Following these studies, a B-cell acute lymphoblastic leukaemia (ALL) humanized mouse model was used to generate CD-19-targeted CAR-T cells from human autologous mature T cells and successfully recapitulated the response seen in patients, demonstrating its efficacy and providing an insight into a potential mechanism behind CRS, with granulocyte macrophage colony-stimulating factor (GM-CSF), a pro-inflammatory cytokine, playing a key role in CRS following CAR-T therapy [68]. Such models can be highly valuable in evaluating the efficacy of various CAR-T cells in the treatment of other malignancies, improving safety and reducing resistance by illustrating the underlying mechanisms of newly established treatments.

#### 2.2.2. CAR-NK Cells

The potential of CAR-NK cells in allogenic cell therapy has emerged in recent years due to the desired properties of these cells such as lower incidence of GvHD, accessibility, and short lifespan in vivo, making them more manageable than CAR-T cell therapy as part of an alternative approach to cancer treatment [69].

Our lab employed our third-generation CAR NK92 cell line, which has been shown to target programmed cell death ligand-1 (PD-L1) and display improved cytotoxicity and apoptosis in various human cell lines that express PD-L1, in the NPC humanized mouse model in order to optimise the use of these cells in treating solid cancers and potentially induce their antitumour activity [70]. Indeed, the injection of HSC-derived primary CAR-NK (CAR-pNK) cells in humanized mice showed the stimulation of the antigen processing and presenting pathways, indicating the activation of the host immune system despite the presence of exhaustive TILs. Furthermore, treatment regimens involving CAR-pNK cells and nivolumab (anti-PD-1 antibody) were designed, and these showed synergistic antitumour responses in the humanized mice, thus demonstrating that CAR-NK therapy can be further pursued for the treatment of NPC and potentially other solid tumours [70].

Another study using genetically engineered cord-blood-derived CAR-NK cells expressing IL-15 and an inducible caspase-9 suicide gene showed improved antitumour toxicity in a CDX lymphoma humanized mouse model. Interestingly, the activation of the suicide gene was able to eliminate the CAR-NK cells in vivo efficiently and rapidly in cases of associated toxicities, and the expression of IL-15 helped in the conservation and proliferation of the stem cell memory T-cell phenotype with an improved antitumour function. These anti-CD19 CAR-NK cells are currently being evaluated in clinical trials, thus establishing the humanized mouse model as a bridge to translational medicine [71].

### 2.3. Improving Cytokine-Based Immunotherapy Using Humanized Mouse Models

Cytokines evoke biological functions in the immune system and can enhance the proliferation and survival of T cells and NK cells upon activation. Cytokines like IL-2 have pleiotropic functions, such as proliferating CD8+ T cells and enhancing the cytotoxic activity, while also stimulating regulatory T cells (Tregs), which are associated with the suppression of antitumour response [72]. Cytokine-mediated immunotherapy using an antitumor cytokine called interferon-alpha 2 (IFN-α2) was the first immunotherapy agent approved by the Food and Drug Administration (FDA) in 1986, and it is understood that the use of humanized mice was extremely limited at the time. IL-2 is one of the few cytokines that has been approved in immunotherapy by the FDA, first for the treatment of metastatic renal cell carcinoma in 1992 and then for the treatment of metastatic melanoma in 1998 [73]. Since these studies were conducted prior to the discovery of fully humanized mice, they showed the survival of human and murine T cells in vitro and in wild-type mice and were translated into clinical testing, where they resulted in dose-related toxicities like fever, chills, malaise, and mild hepatic dysfunction while having no effect on tumour regression [74]. Several other studies followed suit, having similar conclusions [75,76,77].

To improve the efficacy of these cytokine-mediated therapies using humanized mouse models, others have explored the potential of a humanized mouse model of ovarian cancer to study the TME and to test the efficacy of IL-12 targeted immunotherapy. While these studies showed no significant decreases in tumour progression upon treatment with liposome-mediated cytokines, they showed elevations in the levels of IFN-γ in the experimental mice, indicating that T lymphocytes and possibly NK cells are both viable and responsive to IL-12 stimulation and thereby providing evidence of the potential of this treatment [78]. Although this study assessed the safety and efficacy of IL-12 as a potential therapeutic, it only used a partial humanized model (human PDX engrafted into NSG). Subsequently, by using fully humanized mice with a high chimerism of the human immune cells, studies may be able to rule out any harmful effects or the overactivation of the immune system.

### 2.4. Immune Response of Cancer Vaccines in Humanized Mouse Models

Following the coronavirus disease 2019 (COVID-19) pandemic, the importance of vaccines has been highlighted. Scientists have been trying to integrate the theory of vaccinations into the treatment and prevention of cancers. Similar to COVID-19 vaccines, cancer vaccines aim to train the body’s immune system to identify and destroy potentially harmful tumour cells, and can be divided into virus-, peptide-, nucleic-acid-, or cell-based vaccines depending on preparation method. Prophylactic vaccines targeting human papilloma virus and hepatitis B virus (HBV) have gained FDA-approval and are widely implemented in clinics to prevent malignancies associated with viral pathogens [79], while therapeutic vaccines are still underway.

Tumour progression is often accompanied by the presence of somatic mutations. When these mutations occur in protein-coding genes, they introduce non-synonymous polymorphisms which may give rise to novel tumour neoantigens [80]. Tumour antigens recognised by T lymphocytes are essential for ensuring cancer vaccine efficacy and to activate the immune system. These mainly include tumour-associated antigens or tumour-specific antigens. Humanized mouse models can be utilised to screen for such neoantigens and further design optimal targeted vaccines. Multiple dendritic cell (DC) vaccines have been generated and studied in humanized mice. One study compared three DC vaccine formulations using a melanoma antigen recognised by T-cells 1 (MART-1) as an antigen, allowing for the selection of the combination that was most robust and showed an enhanced immune response in a 3-day period [81]. Additionally, a novel mAb (HuMAb006-11) against HBV, which is a commonly known cause of HCC, was developed in a human liver chimeric mouse model. This mAb protected humanized mice from an active HBV infection, inhibited viral entry, and assisted in clearing the virions from circulation, thus establishing its prophylactic effects [82]. These studies highlighted the use of humanized mouse models in proposing novel candidates for future clinical translation and selection of the most optimum therapeutic regimen of cancer vaccines.

### 2.5. Targeted Tumour Lysis by Oncolytic Viruses (OVs) Using Humanized Mice

OV immunotherapy enlists organisms that identify, infect, and lyse tumour cells in the TME, aiming to halt or decrease tumour progression by activating DCs with damage-associated molecular patterns (DAMPs) and tumour antigens while sparing surrounding healthy cells. These organisms could be either genetically engineered or naturally occurring viruses. Their use in clinics is gaining importance, with the first FDA approval for oncolytic vaccines being granted in 2015 [83]. They can be used as single agents or in combination with other therapies and have contributed to tumour cell death and increased the overall efficacy of immunotherapies [83]. However, the clinical manifestations of this therapy are highly variable, demonstrating a need for the humanized mouse model. While the injection of OVs into solid tumours in mouse models has been tested and has showed promising results [84,85], studying the interaction between the virus-colonised tumours and human immune system is highly important in order to understand the optimal mode of action of this therapy, which has emphasised the importance of the humanized mouse model.

A humanized glioma-PDX mouse model has demonstrated not only the efficacy of using OVs, such as CRAd-S-pK7, for malignant gliomas, but also a novel intranasal delivery system using CXCR4-enhanced neural stem cells that leads to the tumour-specific delivery of OVs and extends the survival time of test animals [86]. This study provides an effective strategy for translational oncolytic virotherapy and also gives an insight into the signalling axis that can be further explored to study disease mechanisms.

The interactions of the oncolytic vaccinia virus with tumours were analysed using humanized mice bearing A549 (human lung carcinoma) cell lines. This model demonstrated successful selection, infection, and replication of the virus in the tumours in vivo post administration of the GLV-2b372 oncolytic vaccinia virus strain in the humanized tumour-bearing mice, and further indicated targeted tumour cell lysis. This study elucidated the role played by the human immune system upon activation by the OV as there was an expected rise in the levels of NK cells, which are known to play a role in the first-line defence against malignancies and viral infections [46]. Additional studies have investigated the combination of OVs with ICIs in syngeneic murine models and have demonstrated robust immune-mediated antitumour activity [87], with improved results to be established using humanized mouse models.

### 2.6. Combination Therapy with ICIs

Despite the advantages of immunotherapy, there are some sub-groups that do not respond favourably to a single cancer immunotherapy agent. The treatment of solid cancers has proven to be a challenge with their varied populations, poor tissue penetration, and heterogenous populations, leading to drug resistance mechanisms being developed. For such cases, combining different immunotherapy drugs or administering therapeutic antibodies in conjunction with chemotherapeutics and other modes of treatment has been shown to be more effective in enhancing antitumour efficacy [88]. Clinical trials testing the effects of combination therapy involving two ICIs, anti-CTLA-4 and anti-PD-1, have shown promising results in cancers like melanoma and have already been approved by the FDA for clinical use [89], and other trials using combinations of relatlimab (anti-LAG-3 antibody) and nivolumab are already underway [90]. However, these combinatorial therapies often exhibit adverse immune-related side effects due to a lack of proper understanding of the synergy between the incorporated compounds and their possible overlap in other cellular mechanisms. This has hampered the development of new combination therapies and provides an avenue for the humanized mouse model to be of use. Here we describe some models that have been applied for the validation, safety, and efficacy testing of combination therapies, including but not limited to ICI therapy and ACT therapy.

The NPC humanized mouse model was used to test the efficacy of a combination treatment of nivolumab and ipilimumab and showed results consistent with those seen in donor patients [39]. The association of the Epstein–Barr Virus (EBV) with NPC and other cancers like Burkitt’s Lymphoma, Hodgkin Lymphoma, and NK/T cell lymphoma led to a study using HSC-humanized NRG mice injected intravenously with EBV and targeted with adoptively transferred T cells specific for EBV alone and in combination with nivolumab [91]. The upregulation of checkpoint inhibitors in the study model indicated a strong affinity for combination therapy, which was associated with significantly increased survival and reduced tumour burden in tumour-bearing mice, establishing the importance of combination therapy.

Using the HCC immune system–humanized mouse model, our lab showed that combination therapy in the form of dual and triple therapy could be used to validate anti-HCC effects and side effects in humanized mice. Triple therapy using pembrolizumab, bevacizumab (anti-vascular endothelial growth factor (VEGF)), and the Stat inhibitor C188-9 resulted in an improved combinatorial effect, with the lowest level of proliferation and evidence of reduced angiogenesis, thus significantly increasing the efficacy of the anti-HCC response compared with the results of traditional therapy [48]. Another study using HCC-PDX humanized mouse model demonstrated the anti-HCC effect of anti-GPC3 CAR-T cell therapy [92], providing more evidence on the importance of these models for the testing of novel combinatorial therapeutic strategies.

More recent studies have demonstrated the immunosuppressive and exhaustive phenotypes of ovarian cancers and demonstrated the efficacy of using dual blockade with anti-LAG-3 and anti-PD-1 in murine models with a significantly higher number of T cells in the TME, enhanced CD8+ TIL functions, and a reduced subset of Treg cells [93]. One of the more promising results for treating ovarian cancer was discovered by combining a PD-1/CTLA-4 dual blockade with the adoptive transfer of autologous TILs in humanized PDX-ovarian cancer models. In this study, tumour regression, accompanied by the emergence of memory CD8+ T cells that protected the mice from tumour regrowth, was observed [94].

A prior study using a humanized mouse model engrafted with triple-negative breast cancer (TNBC) cell lines showed the relevance of combination therapy for the treatment of TNBC with nivolumab and OKI-179 (histone deacetylase inhibitor) [95]. This treatment significantly inhibited tumour growth and allowed for the dose modification of nivolumab after a certain period. Hence, it not only provided new therapeutic approaches to TNBC but also validated the efficacy of humanized mouse models in the evaluation of immunotherapy and of combination therapies. Combination therapy has been applied in the humanized oncology model by showing the use of temozolomide (a chemotherapy drug) and ibudliast (a macrophage migration inhibitory factor inhibitor) in extending the survival of mice in a glioblastoma-PDX model [96].

The extensive range of studies using these models for testing and evaluating the different components of immunotherapy proves the impact of these models in advancing the field of personalised medicine and oncology (Table 3). The use of human PBMCs and human HSCs helps not only to create a platform to study and understand the effect of these immunotherapies, but also to discover new targets that could be utilised to develop anticancer agents and provide personalised immune-oncology models.

## 3. Emerging Applications of Humanized Mouse Models

### 3.1. Drug Discovery Models

With advancements in the applications of humanized mouse models, these models have entered the next stage of medical innovations, which includes drug discovery, development, and repurposing, along with personalised medicine and therapy prediction models. Understanding the immune profile of a tumour is vital in screening for new therapeutic targets and potential biomarkers for the treatment of tumours.

A thorough insight into the sequencing profile, based upon extensive experimental work and the differences between tumours from immunodeficient NSG mice and from humanized mice, can provide an increased understanding of which molecular biomarkers and signalling pathways are modified, enhancing drug development. Delineating these differences allows for the discovery of novel targets which can be manipulated to generate novel antitumour drugs. The variation between the TMEs observed in tumours from NSG mice and from humanized mice is a clear indication that tumours are affected by the infiltration of human CD45+ immune cells, which either allow tumours to escape immunosurveillance [40] or halt their progression [70]. Post-PDX engraftment in humanized mice, the immune system triggers an anticancer response, as seen by the presence of proinflammatory cytokines in circulation, but this response is rapidly suppressed as the tumour evolves, as seen by the upregulation of immune-checkpoint biomarkers as compared to the NSG, indicating an immunosuppressive phenotype in the presence of TILs [40,70]. These markers, which are present in the humanized mice and not in the NSG—as seen in the HCC and NPC models—provide targets for drug development and discovery. As such, results from the NPC humanized mouse model indicated the discovery of a novel target and prompted the administration of CAR-NK therapy, demonstrating robust antitumor activity [70]. Importantly, these results show that targets found from examining a transcriptomic profile can successfully be utilised to create antitumour drugs and targeted therapies. Further investigation of the differential expression of mRNA from different treatment regimens can elucidate potential targets and administration sequences to improve antitumour efficacy. Recapitulating the clinical observations in humanized mouse models improves the prediction of drug responses and helps in developing the most optimum therapeutic regimens.

The aforementioned studies demonstrated the use of humanized mouse models in discovering novel druggable targets. Future studies can adapt similar methods to look for new pathways that could be manipulated for treatment purposes. Furthermore, since the antitumour efficacies of PD-1/PD-L1 checkpoint inhibitors and other markers have already been established, studies can now look at using approved and marketed drugs that are not used for cancer treatment but are able to modulate the function of these checkpoint inhibitors, thus repurposing them to improve therapy for solid cancers.

### 3.2. Drug Repurposing Using Humanized Mice

Drug repurposing provides an attractive alternative to the long and expensive development of new chemical entities by bringing new therapeutics at much lower costs using clinically approved non-cancer drugs [97,98]. It involves identifying new uses for approved drugs that are out of the scope of their original medical indications [97]. Using humanized mouse models, we can combine experimental and computational approaches to formulate data-driven repurposing hypotheses and warrant rapid translation to clinics.

Signature matching and comparative analysis: Computational biology has become a crucial aspect of modern science and drug development as it involves the systematic analysis of large amounts of data on gene expression, chemical structure, proteomic and transcriptomic analyses, etc. to produce druggable molecular targets [97]. Experimental science forms the backbone of computational analysis, and together, these methods provide information regarding diseases or targets which can then be manipulated for treatment purposes. Signature mapping, which forms a part of computational biology, is based on comparing the unique features or ‘signatures’ of a drug against those of another drug, disease, or clinical data. These signatures could be transcriptomic, proteomic, or metabolomic data which can be used to make drug–drug or drug–disease comparisons.

One approach towards drug repurposing could be to identify certain pathways and mechanisms that have been manipulated in humanized-PDX mouse models using experimental approaches and comparative analysis to be cross-referenced with drugs that target these pathways for different purposes and are already in the market. In this regard, genetic alterations and mitochondrial protein synthesis have been shown to play roles in the survival of cancer cells, including when these cells are exposed to therapy. Certain antibiotics, like tetracyclines, can be repurposed to inhibit the mitochondrial translational machinery in eukaryotic cells [99,100]. In this regard, humanized mouse models have been utilised to identify vulnerabilities in resistance mechanisms and further target these using FDA-approved and widely used antibiotics. A study tested the repurposing of tetracyclines like tigecycline and doxycycline in a humanized mouse model for melanoma and showed that these drugs could decrease the growth and viability of melanoma cells that had acquired resistance to immune therapies by targeting the mitochondrial translation, with rapid translation to clinic [100]. Further, a humanized mouse model for double-hit B cell lymphomas (DHLs) with the activation of the MYC and BCL2 oncogenes was used to test the repurposing of tigecycline and related tetracycline in anticancer treatment. Tests of these drugs showed the strong antitumoral effects of these antibiotics in combination with venetoclax in CDX and PDX mouse models, along with their potential to cooperate with rituximab (anti-CD20 antibody), thus establishing their usability in combinatorial treatments in DHL [101].

This model could also provide strategies to design combination therapies which could use non-cancer drugs that induce pro-inflammatory responses and ICIs, thus enhancing the antitumour effects of treatments. For example, a drug called liothyronine, used in the treatment of hypothyroidism and as an adjunct treatment for thyroid cancers, has the ability to increase the viability of DCs and potentiate their immunogenicity by stimulating a cytotoxic T-cell response which could potentially work against tumours upon binding to thyroid hormone receptor B [102,103]. Upon studying the compound, it was also found that this drug forms stable complexes with PD-L1 [104], thus presenting a new strategy for the treatment of cancers using liothyronine to inhibit the PD-1/PD-L1 interaction and enhance the capacity of immune cells to fight cancer. While this strategy is currently a hypothesis that has not been validated, it is interesting to note the potential of the compound used in the study as a repurposed drug in the treatment of cancers. Similar strategies can be utilised and optimised using humanized mouse models, starting with finding a unique target that enhances the immunogenicity of cells, then identifying this target in an approved drug and repurposing it for the treatment of cancer.

Further, retrospective clinical analysis using electronic health records can also lead to drug repurposing. Repurposed drugs can be validated using our model, along with having their safety and efficacy checked, all while saving time, labour, and cost. This possibility brings to light a new avenue for the use of our existing humanized models and cements their need in the biomedical research industry.

### 3.3. Uncovering New Disease and Resistance Mechanisms Using Humanized Mouse Models

Since humanized mouse models accurately represent the human immune system, it is possible to survey the effects of tumour-immune interactions and assess the pathways affected in the process, gaining a broader understanding of disease aetiology, resistance mechanisms, and mechanisms of action underlying cancer immunotherapy. This aspect of humanized mouse models has not been explored as much as others and opens new avenues for their utility in the discovery of disease mechanisms.

Understanding disease mechanisms and replicating these in a non-human model is highly important for understanding the possible immunological side effects that could be presented by different biologics. Repurposing existing de-risked compounds and drugs also requires a profound understanding of the mechanisms underlying disease progression and drug resistance before novel sequences can be designed. Insight into the unknown mechanisms of action of new drugs or biologics also allows for accurate safety assessment and designing optimal dose-administration protocols for patients [105].

A humanized mouse model, which was generated by researchers who sought to compare the effects of known immunotoxic biologics like IL-2 and OKT-3 between a humanized mouse model’s predictions and clinical outcomes [105], proved able to replicate in vivo pathological changes and human immune responses more accurately than other models and detailed a competitive dynamic process between inflammation and immunosuppression as was seen in the clinical data. Studying the adverse effects of these biologics led to a deeper understanding of the immunological landscape, thus confidently enabling the use of this technology in order to determine human-specific biologic mechanisms of action [105]. Further, using our humanized mouse model, an in vivo model for a human-specific circadian rhythm of leukocyte trafficking was generated, and the researchers successfully discovered key mechanisms of circadian rhythms by identifying a unique pathway involving p38MAPK/MK, ROS, HIF1α, ARNTL1, and CXCR4 in the regulation of leukocyte trafficking, thus proving the usefulness of this model in uncovering novel disease and regulatory mechanisms [106]. Finally, the BLT humanized mouse model, along with a multiscale 3D tissue imaging approach, was used to study the mechanism of human immunodeficiency virus 1 (HIV-1) dissemination in bone marrow (BM), and this model provided detailed novel structural insights into the virus dissemination process, including on the migration of HIV-1 infected cells to and from the BM to the lymphoid tissue (with the transmission happening via both cell-free and cell-associated mechanisms), and elaborated a route by which macrophages produce, store, and release viruses in tissues [107]. These studies prove that humanized mouse models indeed provide an exceptional opportunity to elucidate unknown disease mechanisms in detail and have led to the application of these models in oncology research.

Given the application of such models in understanding disease mechanisms, the HCC humanized mouse model was used to study the interaction between multiple pathways to illustrate the mechanism of disease progression. This model highlighted the potential of cancer cells to change gene expression in response to immune interaction and detailed a comprehensive analysis of the mechanism underlying HCC progression via the activation of the JAK/STAT pathway in tumour cells by the human immune system, implying that the DAMP/TLR4/AP-1/IL-33 mechanistic pathway was responsible for HCC survival and proliferation [40,48]. This was an important study that paved the way for using humanized mouse models as modes for uncovering disease and treatment mechanisms as compared to simply using them as drug-testing platforms.

It has been known that chronic liver disease and cirrhosis have been major factors in the development of HCC. To this effect, our lab has been studying liver-associated diseases in order to decipher the underlying mechanisms, and its findings could further lead to a better understanding of the pathogenesis of these diseases and the generation of new drug targets. A humanized model of viral-induced liver disease showed the crucial role of GM-CSF in liver fibrosis by reducing the accumulation of intrahepatic tumour necrosis factor-α (TNF-α)-producing CD206+ macrophages and liver fibrosis upon neutralisation of GM-CSF [108]. This may be in part through the production of TNF-α, implying the role of tissue inhibitor of metalloproteinase 1 and transforming growth factor-β in the aforementioned process. The direct involvement of gut-derived microbial products in the mediation of immune activation was revealed in these studies [109,110], thus showcasing the complex multiple-signalling pathways interlaced in this process. It was further found that the M1/M2 paradigm that was based on a limited set of ligands in the immune system [109] might have been insufficient to describe the in vivo function of tissue macrophages when complex multiple signals were involved. This warrants the need for a human-like in vivo model system for identifying these human-specific mechanisms.

Further, despite displaying an exhaustive phenotype in the NPC model, treatment with ICIs did not have a significant effect on tumour growth inhibition, suggesting the possible role of alternate resistance mechanisms involving other checkpoint inhibitors like LAG-3 and/or TIM-3 [39]. The model defined mechanisms of tumour–immune interactions through DAMPs, TLRs, cytokines, and intracellular protein phosphorylation, and showed their association with immune response regulation and tumour progression [39,70]. Collectively, this information elucidates the mechanisms underlying the progression of NPC and provides insights into developing next-generation immunotherapies.

Humanized models also provide information on disease mechanisms taking place at a nuclear level. An acute myeloid leukaemia (AML) model shed light on the key role of NPM1c gene expression in driving the development of AML, with an upregulation of genes involved in the cell cycle and DNA replication, and pointed towards a feedback mechanism between NPM1c, MYC, and the positive regulation of NPM1 transcription [111]. The presence of leukemic stem cells (LSCs) and other genes in turn led to a proof-of-principle [112] study demonstrating the efficacy of a CD123/CD3 bi-specific Fab conjugate in targeting CD123+ LSCs in a T-cell-dependent manner [111]. This study provided further support for the use of humanized mouse models in analysing the mechanisms of disease induction and progression and also the mechanisms of action of immunotherapies.

It is important to note the influence of drug resistance mechanisms on therapy failure and relapse. The isolation of subclones that propagate disease progression from tumour samples in humanized mouse models is essential to delineate the cellular and molecular characteristics that could explain mechanisms of resistance and drug tolerance. One study used a humanized mouse melanoma model wherein a combination of anti-PD-1 antibody with targeted tyrosine kinase inhibitor sunitinib or imatinib resulted in the complete tumour reversion associated with mast cell depletion, identifying both a possible source of resistance for cases where primary resistance to treatment is faced as well as the underlying pathway through which this process may occur [113]. The use of the humanized model for ovarian cancer has also implicated certain factors that could lead to ICI treatment resistance and modulate complex transcriptional changes, including class-I MHC-mediated and ubiquitination-mediated antigen processing pathways [94]. The xenografting of disease samples in humanized mouse models along with sequencing studies has uncovered a subclone of cells in a B-ALL model with increased clonal propagation kinetics and immunophenotypic plasticity that could be the root of drug resistance to chemotherapeutic agents, with the presence of these cells being noted prior to chemotherapy (thus answering the question of whether these functional properties of relapse clones are induced by chemotherapy and elucidating the role of mitochondrial mechanisms in relapse-initiating cells in B-ALL) [114].

Finally, hypoxia is known to be a characteristic of solid tumours and promotes tumour resistance to immune functions and immunotherapy [115]. However, the mechanisms underlying these effects has so far been undetermined. This is particularly evident in TNBC, where, despite having a high number of TILs, the response to immunotherapy has been suboptimal. A study has highlighted the role of hypoxia in T and NK effector cell dysfunction by using humanized mouse models. Through this study, the role of HIF1a and associated epigenetic dependency in the resistance mechanisms of PD-1 blockade have been demonstrated. Using pembrolizumab effectively repairs immune cell dysfunction with increased expressions of IFN-ɣ and TNF-α and intratumoral human CD8+ and NK cells [116]. These findings have revealed a mechanism to explain resistance to treatment and highlight the underlying immunotherapy and disease progression. Taken together, all these studies illustrate the achievements arising from the utilisation of humanized mouse model technology in areas ranging from disease modelling to drug development and the discovery of new mechanisms.

### 3.4. Identifying Biomarkers of ICIs to Improve Predictability of Drug Responses

The use of biomarkers is being recognised as an extremely important step in designing treatment strategies to identify patients who are most likely to respond to a certain treatment and reduce exposure and risk to those that are least likely to respond, or have adverse reactions, to the same treatment. Proteomic profiling methods and comparative analyses allow for the screening of various samples to discover protein or genetic markers that could provide better diagnosis, prognosis, and monitoring of disease spread in the case of metastasis. The vast heterogeneity of solid tumours again serves as a deterrent in discovering new biomarkers, and the process of designing and executing such studies remains a challenge [117]. The use of humanized mouse models in modelling such studies and identifying proteins in circulations at various time periods provides an excellent approach to overcome current challenges. Briefly, a study using a partially humanized breast cancer PDX model summarised the advantages of this model in identifying novel biomarkers specific for detecting cells with metastatic potential. By doing a comparative analysis of human-specific gene expressions from the BM of humanized mice that developed metastasis, xenografted mice that did not show distant metastasis, and non-transplanted control mice gene transcripts specific to epithelial-to-mesenchymal transition (EMT), aggressive clinical phenotype and metastasis were identified [118]. This model also helped identify markers that could distinguish between healthy volunteers and breast cancer patients with early metastatic potential, thus serving as a valuable tool for identifying predictive and prognostic biomarkers [118].

While the importance of positive biomarkers is well-known, it is essential to note the complementary role of negative predictive biomarkers. These markers identify tumours resistant to therapy while also providing insight into the mechanisms underlying immune evasion during tumorigenesis. An in vivo humanized mouse model was established to study the dynamic temporal interactions between alternate promoter use and the human immune system [119]. Alternate promoter burden (APB) is a key regulator of gene expression and an important factor for transcriptome diversity. It may influence tissue specificity and protein translation and functions [120]. By analysing stomach adenocarcinomas and comparing tumour growth kinetics between humanized and immunodeficient mice, the predictive nature of APB specific to ICI treatment in the humanized mice was identified, thus qualifying a selection of patients with gastric cancer for immunotherapy by classifying them into a “likely responders group” with positive biomarkers, an “unlikely responders group” consisting of APB^high^ patients, and a “possible responders group” consisting of the remainder of the patients who may demonstrate only moderate benefits from immunotherapy with possible resistance [119].

Thus, humanized mouse models have become indispensable tools in translational research and have portrayed their utility in the discovery of biomarkers.

## 4. Next-Generation Humanized Mice Models

Humanized mouse models are not without challenges. Efforts have been made to tackle the current limitations involved in the incorporation of humanized mice and provide models that have higher and more complete human immune system reconstitution.

Cytokines are known to play a key role in the development and survival of NK cells, DCs, and other myeloid cells [121,122]. It is also known that cytokines are species-specific, as demonstrated by the absence of effect of many mouse cytokines on human immune cells and their precursors [121,122]. Thus, the impaired NK cell and myeloid cell expansion in the current humanized mouse models could be attributed to the lack of human cytokines. Improving the human cytokine profile by inducing transgene expression was theorised to be a method of generating humanized mice with higher NK-cell reconstitutions and improved myeloid-cell reconstitutions and thus having more complete immune reconstitutions. Many groups have also attempted to improve the cytokine profile to improve the reconstitution and functions of human immune cells in humanized mice, some of whose methods are listed in the table below (Table 4).

### 4.1. Hydrodynamic Injection Method for Improved Cytokine Model

Hydrodynamic gene delivery is a widely used method of delivering nucleic acid and producing high levels of transgene expression in mice [132]. The pressurised tail-vein injection of a large volume of DNA solution (8–10% of body weight) in a short duration (6–8 s) causes liver damage, permeabilising the capillary endothelium and allowing the uptake of DNA by hepatocytes [122,132].

While several reports have stated the effects of adverse drug reactions in humans, many of these effects fail to appear in mice due to the fact that they have different analogues of receptors on mast cells, which are critical for these reactions [13,14,15,16]. Through the hydrodynamic injection of GM-CSF and IL-3, we were able to generate humanized mice that were highly enriched in mast cells at various body sites [122]. When extracting cultures from the BM of these mice, we were able to replicate the degranulation response seen in clinical studies upon exposure to drugs known to cause adverse reactions in humans [123]. The hydrodynamic tail-vein injection of plasmid DNA encoding human erythropoietin and IL-3 was also able to significantly enhance the reconstitution of erythrocytes in the humanized mouse model, thus proposing that the utilised method was fit for the overall improvement of human chimerism in mice [133]. This finding shows the significant value of these improved humanized models and prompts their further utility in various studies.

### 4.2. Transgenic Mice Using Pronuclear Injection

Another commonly used method of generating transgenic humanized mouse models includes pronuclear injection, wherein the transgenic DNA construct is physically microinjected into the pronucleus of a fertilised egg [128,134].

Using pronuclear injection on human stem and progenitor cell (HSPC)-humanized SGM3 mice (NSG-SGM3), a triple transgenic model that constitutively produced stem cell factor (SCF), GM-CSF, and IL-3 and successfully recapitulated both CRS characteristics as well as neurotoxicity in mice was generated. Through this model, it was found that human circulating pro-inflammatory monocytes are a major source of IL-1 and IL-6 [127], and this discovery presented novel cellular and molecular targets for therapy. However, while tocilizumab (IL-6 receptor antagonist) prevented only CRS, anakinra (IL-1 receptor antagonist) was able to control both CRS and neurotoxicity, resulting in extended leukaemia-free survival [127]. The cited studies have been significant in emphasizing the importance of improved humanized mouse models for the enhanced and more accurate safety profiling of CAR-T cell therapy—an emphasis that can be extended to other adoptive cell therapies as well.

### 4.3. Knock-in Models of Improved Humanized Mouse Models

Understanding the different roles played by specific cytokines in the development of the human monocyte and macrophage compartment, a new humanized mouse model was generated using immunodeficient Rag2^−/−^Il2rg^−/−^ mice wherein a combination of human macrophage colony stimulating factor (M-CSF), human IL-3, and GM-CSF and human thrombopoietin (TPO) were knocked into the respective mouse loci [131]. These mice were called MI(S)TRG. MISTRG mice bear a BAC transgene encoding human SIRPα that suppresses the phagocytosis of human CD47-expressing cells, thus enabling highly permissive human haematopoiesis [131].

The described model further supports the application of humanized mice models in translational science and is popularly used in research. Another study modelled human multiple myeloma plasma cell tumours using the MISTRG models with an additional knock-in allele to express human IL-6 and recapitulated the entire genetic diversity of the primary tumour, witnessing the improved growth of non-malignant cells from the primary tumour over other models, likely due to enhanced human haematopoiesis, even in the absence of human foetal tissue [135]. Using this model will help further advance the development of personalised therapies and understand the heterogeneity of human tumours.

## 5. Limitations

Despite the advancements made in humanized mice models, there are still certain practical limitations that prevent the current models from being fully identical with human biological systems. One of the major limitations in humanized PDX models or cancer models is the mismatch between the cells obtained from the HSC donor and the human tumour, thus making them only partially HLA-matched. In the future, we hope to establish fully HLA-matched and personalised humanized mouse models, hence negating the possible mismatching of cells and establishing a fully comparable and autologous model system. This end is now being pursued, with studies establishing models by generating cancer cell lines from donor-specific human PBMCs. While this might offer a possible method to study blood cancers currently, there is still a long way to go before this model can be optimised for solid cancers. Further, induced pluripotent stem cells (iPSCs) may be used to create patient-matched mice for studies including those on infectious diseases and hepatic virology. However, iPSCs have low in vivo repopulating ability, and this makes the formation of a fully functional immune system challenging [4].

Another concern with these models has been the lack of HLA class-I and class-II expression in the mouse thymus, which are required for the selection of T cells post the engraftment of human cells. To address this issue, transgenic HLA mice have been humanized by engrafting human HSCs, autologous marrow stroma, and thymic tissue in them to create humanized transgenic mice [136,137]. These MHC molecules successfully interact with T cells in the thymus and select the ones expressing various antigen receptors, with the costimulatory molecules considering the HLA as self and interacting with them [137]. Thus, optimising the combination of transgenic and humanized mouse models could provide a fundamentally strong system to recapitulate the human system and generate precise analyses.

## 6. Future Directions

So far, we have known humanized mice as mice harbouring human immune systems, but the incorporation of other human biological systems is also essential to support more human-specific studies on topics such as infectious diseases that target other organs and form cross-networks with the immune system. With respect to drug development, the liver is known to play an important role in drug metabolism and excretion along with being a frequent target for toxic manifestations of adverse drug effects [138]. In addition, the liver harbours the resident macrophages, known as Kupffer cells, and is known to be important for producing cytokines and complements that support human immune cell development and functions.

This knowledge has led to the discovery and generation of dual-humanized mice, the models of which consist of human liver cells and human immune systems reconstituted in immunodeficient mice. While this dual humanisation has been previously attempted using different strains of genetically modified immunodeficient mice and foetal or mature hepatocytes, the hepatic repopulation levels have been low [139,140]. Recent developments in FRG knockout immunodeficient mice have led to the generation of a stable liver chimeric mouse model [141]. This model has been widely used to study hepatotropic infections, drug metabolism and hepatotoxicity, and drug testing and has provided novel insights into the pathogenesis of infectious diseases such as malaria [142], HBV [33,141], and hepatitis C virus (HCV) infection [143].

Our lab is one of the first in Southeast Asia to demonstrate the successful engraftment of human foetal liver-derived HSCs and hepatoblasts in NSG mice for the study of infectious liver-associated diseases [144]. Using the FRG(N) strain has led to the creation of models with high degrees of liver humanisation, involving mature hepatocytes and increased haematopoietic engraftment [145]. The HIL model, with human liver cells and a matched human immune system in NSG mice, showed high efficiency in replicating HCV infection as seen in patients and also reproduced the antiviral effects of human IFNα2a therapy [146]. These models further help to dissect the mechanisms of human immune responses to hepatitis virus infection and provide an accurate platform for testing new therapeutics and vaccines. Therefore, dual humanisation models provide a glimpse into the future of humanized mice, have the potential to better assess human-specific immune responses, and reveal a path to further improvement.

Moving further, since many research groups are now experimenting with the generation of dual humanized mice using the FRG strain, the way forward is to establish triple humanisation with a human liver, immune system, and specific organ, with the assumption that these improved models will better recapitulate the efficacy and safety profiles observed in clinical settings. These improved models will provide a clearer understanding of human diseases and be the last pieces of the puzzle that is a human-specific model system.

Recently, a new piece of legislation has passed stating that the FDA no longer requires animal testing for the approval of new drugs. In the past, FDA approval for a new drug required safety and toxicity testing in one rodent species and one non-rodent species prior to human trials. Despite these preclinical tests, more than 90% of drugs showed toxic side effects in patients and clinical trials [147], questioning the authenticity of animal experiments. The inaccuracy of these studies led to a large number of animals being used to provide reproducible and practical results. In this regard, the efficacy of humanized mouse models and their accurate representations of the human system set humanized mice apart from the presently used rodent and non-rodent species. While bioengineers are pushing for the use of organ chip technology, 3D organoids, and stem cells to replace animal testing, there is no certainty regarding whether the mechanisms underlying their proposed drugs would be implicated in the functioning of other organs. Moreover, once the complex interactions between the diverse cells of the immune system come in play, these drugs could have different effects than those observed on organ or stem cell platforms. For these reasons, while it is prudent to realise the shortcomings of animal experiments, there is an increased need to implement the use of humanized mouse model technology to replace them as compared to the usage of other suggested platforms. The numerous examples cited in this review are clear indications of the validity of these models and highlight their utility in predicting the outcomes of drug effects when compared to the utilities of wild-type animal models. With all these benefits, it would be highly advantageous in the future to make the use of humanized mouse model technology a possible requirement to gain FDA approval in the process of drug development, thus eliminating the need for wild-type animal testing and saving immense amounts of time, money, and lives.

## 7. Conclusions

As we have seen, the evolution of mouse models has been a rather long journey with many recent advancements. While there are still certain limitations to the current models, it has been proven that these models are indispensable for preclinical testing and provide major platforms for the development of novel vaccines and new therapeutic strategies for previously incurable diseases. Additionally, our own model relatively reduces the burden of cost and time in studies while providing data on tumours using multiple genetically identical mice with human immune systems, innate inconsistency. Such models continue to further our understanding of the human immune system and its disorders and will likely provide further insight into developing the field of regenerative and personalised medicine.

Taken together, humanized mouse models are important preclinical tools with great utility in scientific translation. With the advantage of having human immune systems, these mouse models serve to elucidate drug response, safety, and pharmacological studies of different interventions, and target identification for cancer drug development. Ultimately, these efforts seek to achieve improvements in clinical outcomes for patients. Encouragingly, these applications are merely in the field of oncology, showing the vast, possibly untapped potential of humanized models as they can be employed in almost every field of medical science.

## Figures and Tables

**Figure 1 pharmaceutics-15-01600-f001:**
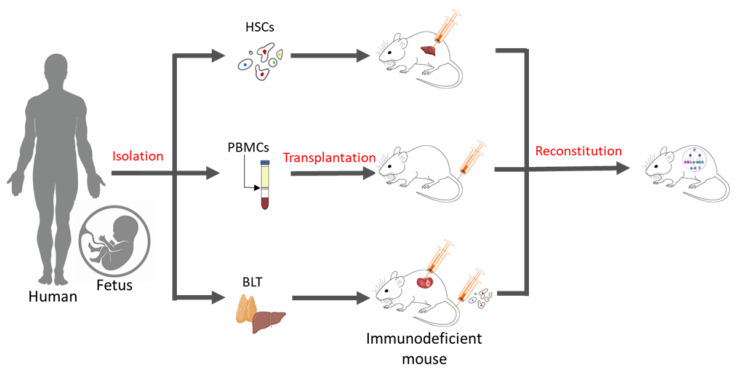
Methods of generating humanized mice. Humanized mouse models may be generated in three ways: (**a**) By the intra-hepatic injection of adult or foetal HSCs in immunodeficient mice, (**b**) by administering human PBMCs via tail-vein injections, and (**c**) through the simultaneous transplantation of human foetal liver and thymus tissue under kidney capsules and the intravenous injection of human CD34+ HSCs in immunodeficient mice.

**Figure 2 pharmaceutics-15-01600-f002:**
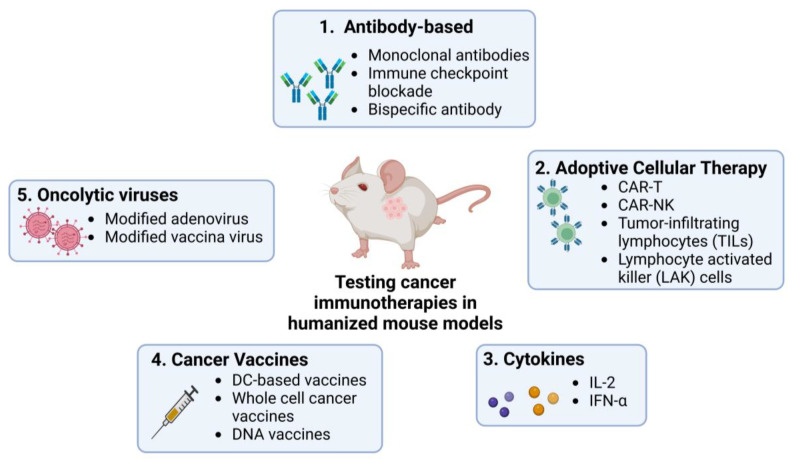
Use of humanized mouse models to test current immunotherapies in cancer research. Many immunotherapies, including ICIs, ACT, cytokine-based therapies, and therapies using cancer vaccines and oncolytic viruses, are currently being studied using humanized mouse models. These models provide platforms for testing the efficacy and safety of the commonly known therapies mentioned in the figure in human-like systems. Created by BioRender.com.

**Figure 3 pharmaceutics-15-01600-f003:**
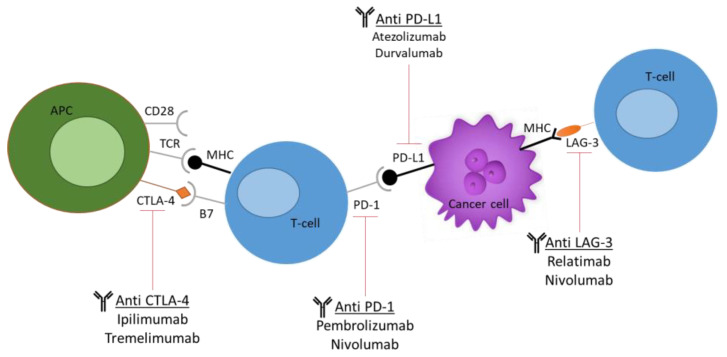
Current immune checkpoint inhibitors (ICIs) used in cancer research. The most commonly used ICIs, including anti-PD-1, anti-PD-L1, anti-CTLA-4, and anti-LAG-3. The figure demonstrates the molecular and cellular interactions between the T cells and cancer cells and the immune response activation upon antibody administration.

**Table 1 pharmaceutics-15-01600-t001:** Most commonly used immunodeficient mice models. Much progress has been made in the development of immunodeficient models for the engraftment of human immune cells. While earlier models also supported the engraftment of PBMCs and HSCs, the reconstitution and chimerism in these models were low, possibly due to the effects of residual murine cells. This table summarises the most commonly used immunodeficient models for different applications along with their distinct characteristics.

Models	Mutations	Advantages	Disadvantages	Refs.
SCID	Protein kinase DNA-activated catalytic peptide gene (*Prkdc*)	Lack humoral and cell-mediated immunity Absence of B and T cells	Leakiness due to age Reduced lifespan Presence of murine NK cells Complementation of activity	[20,23]
NOD/SCID	Signal regulatory protein alpha (*Sirpa*)	High affinity to human CD47Tolerance of host macrophages to the human cellsImmunological multi-dysfunction, including defective NK cell activitySupport higher levels of human engraftments	Residual NK cell activity, and some other innate compounds of the immune system, fail to develop mature monocytes, which eventually become thymic lymphomas with age	[24,25,26,27,28,29]
BRG Rag1/2	Recombination-activating gene 1 (*Rag1*) or 2 (*Rag2*)	No leakiness Absence of functional B and T cellsLonger lifespans	Limited lymphoid reconstitution Residual NK cell activity	[30]
NOG NSG	Interleukin-2 receptor gamma chain (*IL2r*γ) null phenotype and Prkdc mutation (NOD/SCID-IL2Rγ^−/−^)	Absence of functional receptors for cytokines like IL-2 and IL-7 Could hamper the development of host NK cells Highest rate of human cell engraftments Longer lifespans	Weak human myeloid reconstitution Lack of human thymus tissue Radiation sensitivity	[21,22,28,31]
NRG	RAG1/2 null mutation and Interleukin-2 receptor gamma chain (*IL2r*γ) null (NOD/RAG1/2^−/−^- IL2Rγ^−/−^)	Improved myeloid engraftmentCan tolerate chemotherapy at higher doses	Higher radiation dose for preconditioning Lack of human thymus tissue Possible background activity	[32]
FRG	Fah/Rag2/IL-2rγ	Triple knockout Expand human hepatocytes robustly Gene and cell therapy	Remnant mouse hepatocytes Background activity	[33,34]

**Table 3 pharmaceutics-15-01600-t003:** The use of humanized mouse models in different immunotherapies, Summarising their applications and safety and efficacy profiles along with indications on how they can be improved to overcome their current limitations and provide an enhanced platform for drug testing and development.

Immunotherapy	Humanized Mouse Model Application	Safety & Efficacy	Further Modifications	Limitations
Antibody-based	Help in discovering new targets for ICIs	Improved compared to wild type mice	Immune-transgenic models	Undefined resistance mechanisms
Adoptive Cell Therapy	Hints towards possible mechanisms underlying CRS	Higher safety and efficacy standards	Improved NK cell model for cytotoxic activity	CRS and neurotoxicity, limited effect on solid tumours
Cytokine	Gaining popularity to prevent tumour growth	Potential to improve	Fully humanized models with improved cytokine transgenic models	Lack of specificity
Cancer vaccines	Prophylactic and therapeutic vaccines are being established using this model	Dose escalation studies provide improved safety and efficacy	Dual humanized mouse models for specific viral cancers	Scarcity of suitable neoantigens
Oncolytic viruses	Discovering novel delivery systems and targeting tumour lysis	Improved safety and efficacy when translated to clinic	Dual humanized mouse models	Need for improved delivery strategies, tumour heterogeneity

**Table 4 pharmaceutics-15-01600-t004:** Methods used to develop next-generation humanized mouse models. An overview of some of the methods used for improving cytokine levels and immune cell reconstitution in humanized mouse models, highlighting the pros and cons of using each model and the specific applications of each model.

	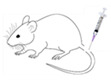	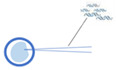	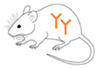
Method of Delivery	Hydrodynamic Injection	Pronuclear Injection	Knock-In
Cytokines/Modifications	IL-1B, IL-2, IL-7, GM-CSF, Flt-3L	NSG-SGM3: SCF, GM-CSF and IL-3	Hu-IL15: Human IL-15	SRG-15: Human IL-15 and SIRPα	MITRG/MISTRG: Human M-CSF, human IL-3 and GM-CSF, and human TPO
Key Features	Improved NK cells Proliferation of myeloid cellsIn vivo transfection of hepatocytes	Development of abundant and functional mast cells	Enhanced human NK cell development	Functional human NK cells	Multilineage differentiation of human B, T, and myeloid cells supports development of NK cells
Advantages	Easy to implement Time-savingImproved antigen-specific antibody responses	High level human haematopoietic engraftment initially	Improved capacity for cytotoxic activity	Simple use, no cytokine administration	No radiation preconditioningLong-term haematopoiesisClosely resembles diversity and cell populations seen in humans
Disadvantages	Possible damage to target tissue due to high and rapid pressure	Loss of stemness and functional properties over timeLong-term effects of irradiation	Maturation differences between NK cell subset from humans and HSC-engrafted NSG-Tg (Hu-IL15) mice	High concentration of IL-15 could result in impaired functionality	Thrombocytopenia, hemophagocytosis, short lifespan, anaemia
Application	Liver-associated,Infectious diseases, immune responses, adverse drug reactions	Systemic anaphylaxis	Study of PDX tumours	Studying human tissue-resident immune cells, combination therapy protocols	Studying haematological malignancies
References	[122,123,124,125,126]	[127,128,129]	[129,130]	[129]	[129,131]

## Data Availability

Not applicable.

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
