# Peer review of "Emerging Preclinical Applications of Humanized Mouse Models in the Discovery and Validation of Novel Immunotherapeutics and Their Mechanisms of Action for Improved Cancer Treatment"

_pharmaceutics, 2023, doi:10.3390/pharmaceutics15061600_

Round 1

Reviewer 1 Report

In this review, authors summarized the knowledge about humanized mouse models’ establishment, and how to use humanized mouse models in pre-clinical research to validate and evaluate the efficacy and safety of immunotherapeutic strategy. This paper may benefit and provide guidelines for the researchers in the area of immunotherapy. I just have few questions below.

1.     Table 2, as Anti-PD-L1 antibody is also a wildly use immune checkpoint inhibitor, can authors add some examples here in which Anti-PD-L1 had been applied?

2.     Figure 2, as bispecific antibody also has been used in humanized mouse model for cancer immunotherapy research, can authors add this immunotherapeutic strategy in the figure?

This paper is well written, so just minor editing of English language required

Author Response

The authors would like to thank the reviewer for their feedback and appreciation of our paper.

Major points: 

1. Table 2, as Anti-PD-L1 antibody is also a wildly use immune checkpoint inhibitor, can authors add some examples here in which Anti-PD-L1 had been applied?

Response: Understanding the importance of Anti-PD-L1 antibody and their use as ICIs, we have included a few relevant examples of their testing in humanized mice and shown that these mice provide a highly effective platform for testing the safety and efficacy of anti-PD-L1 antibody prior to their use in clinics.

2. Figure 2, as bispecific antibody also has been used in humanized mouse model for cancer immunotherapy research, can authors add this immunotherapeutic strategy in the figure?

Response: The figure has been amended to include bispecific antibody in the ICI-based immunotherapy research.

Reviewer 2 Report

This is a well-written review of the pre-clinical applications of mumanized mouse models in the discovery and validation of novel immunotherapeutics and their mechanisms of action for improved cancer treatment. This review describes the humanized mouse models, highlights the challenges and recent advances in these models for targeted drug discovery and validates the therapeutic strategies in cancer treatment. It also describes the potential of these models in the for uncovering novel disease mechanisms.

The introduction is well written and gives the rationale for the review. The applications of humanized mice in oncology is well researched and the commonly used immunodeficient mice models are presented by the authors. The limitations of the humanized mice models  and current advances in overcoming the limitations are discussed. 

The tables and figures are well designed and informative.

Again, I believe the review is well written and interesting to read, and there is no additional comment from this reviewer.

Best of luck!

Author Response

The authors would like to thank the reviewer for the positive feedback and appreciation of our manuscript and for acknowledging the importance of humanized mouse models for immunotherapy and oncology research.

Reviewer 3 Report

This review entitled "Emerging Pre-Clinical Applications of Humanized Mouse  Models in the discovery and validation of novel immunothera- peutics and their mechanisms of action for improved cancer treatment." by Karnik I. et al., summurized application of humanized mouse models of new immunothrapies in cancer treatment. This is very important in this field. But some corrections may be required.  In section 2, line 178, it was better to describe the relationship between HLA-rescrict and immunoresponses more detail. In addition, it was better to show the molecular and cellular interactions of immunoresponses figures.  In references, some very old references are inclueded. It was better to add recent references. 

It was very good.

Author Response

The authors would like to thank the reviewer for their feedback and comments on how to improve the manuscript. Below please find our point-to-point response to address the questions:

Major points:

  1. In section 2, line 178, it was better to describe the relationship between HLA-rescrict and immunoresponses more detail.

Response: Thanks for the suggestions. In the revised manuscript, we discussed the relationship between HLA-restriction and immunoresponses for better understanding the interactions between mouse and human immune systems (Line 178-183).

  1. In addition, it was better to show the molecular and cellular interactions of immunoresponses figures. 

Response: To give the readers a better understanding of the immunoresponses, we have included a figure (Figure 3) showing the molecular and cellular interactions between the T cells and the cancer cells with respect to ICI-based therapy detailing the different ICIs used in research currently.

  1. In references, some very old references are included. It was better to add recent references. 

Response: We have revised some of our references to provide a more up-to-date information on the applications of humanized mouse models in studying immunotherapy, especially ICIs targeting PD-L1 as anti PD-L1 antibody is also a widely used ICI which has been gaining more popularity in recent years.

  1. Lin S, Huang G, Cheng L, Li Z, Xiao Y, Deng Q, Jiang Y, Li B, Lin S, Wang S, Wu Q, Yao H, Cao S, Li Y, Liu P, Wei W, Pei D, Yao Y, Wen Z, Zhang X, Wu Y, Zhang Z, Cui S, Sun X, Qian X, Li P. Establishment of peripheral blood mononuclear cell-derived humanized lung cancer mouse models for studying efficacy of PD-L1/PD-1 targeted immunotherapy. MAbs. 2018 Nov-Dec;10(8):1301-1311. doi: 10.1080/19420862.2018.1518948.
  2. Li Y, Carpenito C, Wang G, Surguladze D, Forest A, Malabunga M, Murphy M, Zhang Y, Sonyi A, Chin D, Burtrum D, Inigo I, Pennello A, Shen L, Malherbe L, Chen X, Hall G, Haidar JN, Ludwig DL, Novosiadly RD, Kalos M. Discovery and preclinical characterization of the antagonist anti-PD-L1 monoclonal antibody LY3300054. J Immunother Cancer. 2018 Apr 30;6(1):31. doi: 10.1186/s40425-018-0329-7. Erratum in: J Immunother Cancer. 2018 Jun 4;6(1):45.